# Large-Sample Genome-Wide Association Study of Resistance to Retained Placenta in U.S. Holstein Cows

**DOI:** 10.3390/ijms25105551

**Published:** 2024-05-20

**Authors:** Dzianis Prakapenka, Zuoxiang Liang, Hafedh B. Zaabza, Paul M. VanRaden, Curtis P. Van Tassell, Yang Da

**Affiliations:** 1Department of Animal Science, University of Minnesota, Saint Paul, MN 55108, USA; 2Animal Genomics and Improvement Laboratory, Agricultural Research Service, United States Department of Agriculture, Beltsville, MD 20705, USA

**Keywords:** retained placenta, GWAS, SNP, additive effect, Holstein

## Abstract

A genome-wide association study of resistance to retained placenta (RETP) using 632,212 Holstein cows and 74,747 SNPs identified 200 additive effects with *p*-values < 10^−8^ on thirteen chromosomes but no dominance effect was statistically significant. The regions of 87.61–88.74 Mb of Chr09 about 1.13 Mb in size had the most significant effect in *LOC112448080* and other highly significant effects in *CCDC170* and *ESR1*, and in or near *RMND1* and *AKAP12*. Four non-*ESR1* genes in this region were reported to be involved in *ESR1* fusions in humans. Chr23 had the largest number of significant effects that peaked in *SLC17A1*, which was involved in urate metabolism and transport that could contribute to kidney disease. The *PKHD1* gene contained seven significant effects and was downstream of another six significant effects. The *ACOT13* gene also had a highly significant effect. Both *PKHD1* and *ACOT13* were associated with kidney disease. Another highly significant effect was upstream of *BOLA-DQA2*. The *KITLG* gene of Chr05 that acts in utero in germ cell and neural cell development, and hematopoiesis was upstream of a highly significant effect, contained a significant effect, and was between another two significant effects. The results of this study provided a new understanding of genetic factors underlying RETP in U.S. Holstein cows.

## 1. Introduction

Retained placenta in dairy cattle refers to the failure of timely separation of the placenta from the dam after calving. This disease creates a number of problems in health and fertility including inflammation, fever, decreased milk yield, longer calving intervals, higher incidence of metritis and lower conception rate [1]. In U.S. Holstein cows, retained placenta has an incidence rate of 3.6%, and is a low-heritability trait with 1% heritability [2]. In humans, retained placenta was found to have inherited risk where the mother’s having retained placenta increased the risk of this disease in the next generation [3]. Only limited research was available on the association between retained placenta and genetic variants. A genome-wide association study (GWAS) using U.S. Holstein bulls and single nucleotide polymorphism (SNP) markers included retained placenta but found no significant effects [4]. Another study on retained placenta using Canadian Holstein bulls identified several chromosome regions with higher heritability estimates than in other chromosome regions [5]. Starting in 2018, national genetic and genomic evaluations for resistance to retained placenta (RETP) started for U.S. Holstein cattle [6] and have since accumulated a large sample of Holstein cows with RETP phenotypic observations and genotypes of genome-wide SNPs, providing a unique opportunity with unprecedented statistical power for identifying genetic variants associated with RETP. Using this large sample, this study aimed to identify genetic variants and chromosome regions affecting RETP in U.S. Holstein cows using a GWAS approach.

## 2. Results and Discussion

### 2.1. Overview of GWAS Results

The GWAS of RETP using 614,035 first-lactation Holstein cows and 74,747 SNPs identified 200 additive effects and no dominance effects with log_10_(1/p) > 8 (Figure 1a, Appendix A). The 200 significant additive effects were distributed on twelve chromosomes (Figure 1a): 5, 6, 7, 9, 10, 15, 17, 19, 24, 25, 29 and 31, where Chr31 is the X-Y nonrecombining region of the X chromosome. Allelic effects of the 200 SNPs (Figure 1b) showed that negative allelic effects had larger effect sizes than the positive effect sizes for most chromosomes. The average of the negative allelic effects was −0.089 and the average of the positive allelic effects was 0.077. Chr05 had the most positive allelic effect in *C5H12orf75*, and the X-Y nonrecombining region of the X chromosome (Chr31) had the most negative allelic effect between *MAGED1* and *MAGED4B*. The detailed descriptions of candidate genes of the significant effects will use gene symbols for most genes and the full gene names are provided in Appendix A.

### 2.2. Additive Effects of Chr09

Chr09 had twenty-three significant effects including the #1 effect in *LOC112448080*, #2 upstream of *AKAP12*, #3 in *CCDC170*, and #5 in *RMND1* (Table 1 and Appendix A, Figure 2a). These effects were in the 87.61–88.73 Mb region about 1.12 Mb in size. An interesting aspect of this region was the estrogen receptor 1 (*ESR1*) gene fusions observed in humans. This region had *PLEKHG1*, *MTHFD1L*, *AKAP12*, *RMND1*, and *CCDC170* genes upstream of *ESR1* (Table 1), and four of these five upstream genes were involved in *ESR1* fusions associated with human breast cancer, *ESR1-CCDC170*, *ESR1-AKAP12*, *ESR1-MTHFD1,* and *ESR1-PLEKHG1* fusions [7,8,9,10]. In addition, *ESR1* had gene fusions with multiple other genes [7]. These human *ESR1* fusions indicated the potential involvement of *ESR1* fusions in RETP of Holstein cows. The protein encoded by *ESR1* regulates the transcription of many estrogen-inducible genes that play a role in growth, metabolism, sexual development, gestation, and other reproductive functions; and is expressed in many non-reproductive tissues [11]. An SNP in *ESR1* had a significant dominance effect on daughter pregnancy rate in Holstein cows [12]. The well-documented *ESR1* functions including *ESR1* fusions pointed to the possible involvement of *ESR1* in the significant effects of RETP in the 87.61–88.73 Mb region of Chr09.

### 2.3. Additive Effects of Chr23

Chr23 had fifty-two significant effects, the largest number of significant effects among all chromosomes. These effects except one were distributed in the 21.29–35.47 Mb region, about 14 Mb in size (Appendix A), but the most interesting region was the 23.70–33.08 Mb region about 9.4 Mb in size due to the genes potentially contributing to or known to be associated with kidney disease (Table 2, Figure 2b). The most significant effect of this chromosome (#4 overall) was in *SLC17A1*, followed by SNPs in or near *MRPS18B, PPT2*, and the region of *ELOVL5* to *BOLA-DQA2* (Table 2). *SLC17A1* with the most significant effect was involved in urate metabolic process and urate transport [13] and elevated serum urate concentrations could contribute to kidney disease [14]. The *PKHD1* gene had six significant effects and was downstream of another five significant effects (Appendix A). This gene was associated with a severe form of polycystic kidney disease named autosomal recessive polycystic kidney disease (ARPKD) that presents primarily in infancy and childhood [15,16]. The *ACOT13* gene had a significant effect (#16 overall) and was also reported as a candidate gene for ARPKD kidney disease [17]. It was interesting that the two genes known to be associated with kidney disease, *PKHD1* and *ACOT13*, were near the two ends of the Chr23 region with significant RETP effects (Table 2, Figure 2b). *BOLA-DQA2* is a bovine MHC class II gene [18] and MHC class II molecules are critical for the initiation of the antigen-specific immune response [19]. The last significant effect at the very downstream end of the Chr23 region with significant RETP effects was between *PRL* and *HDGFL1*, where *PRL* is the prolactin gene and is essential for lactation [20].

### 2.4. Additive Effects of Chr05 and Chr17

Chr05 had thirty-five significant additive effects (Appendix A). The most significant effect of Chr05 (#17 overall) was at 27,005,657 bp between the *KRT18* and *KRT8* genes, 3394 bp downstream of *KRT18* and 34,610 bp upstream of *KRT8*, noting that neither *KRT18* nor *KRT8* had any SNP inside the gene. Gene Ontology (GO) analysis showed that *KRT8* was involved in embryonic placenta development (Appendix A), and this biological function could be directly relevant to retained placenta. A SNP upstream of *KRT18* (in *EIF4B-KRT18*) at 26,984,066 bp was also significant (#97 overall). It should be noted that another SNP in *EIF4B-KRT18* at 26,964,045 bp had a sharply negative recessive genotype for age at first calving (AFC) [21]. The effects in or near *KITLG* were also interesting due to *KITLG*’s known biological functions. *KITLG* was upstream of the second-most significant effect of Chr05 (#25 overall), contained a significant effect and was between another two significant effects (Table 3, Figure 3a). The *KITLG* gene encodes the ligand of the tyrosine-kinase receptor encoded by the KIT locus, and this ligand is a pleiotropic factor that acts in utero in germ cell and neural cell development, and hematopoiesis [22]. The ‘in utero’ biological functions of *KITLG* could be directly relevant to RETP. The *POC1B* gene had the third most significant effect of Chr05 (#28 overall). This gene has an important role in basal body and cilia formation [23]. 

Chr17 had fifty-one significant effects (Appendix A). The most significant effect of Chr17 (#6 overall) was in *LOC112442091*, and the second-most significant effect of Chr17 (#16 overall) was between *LOC112441992* and *LOC112442089*. All the above three genes had unknown biological functions. The third-most significant effect of Chr17 was between *ANAPC10* and *HHIP*. A 388,434 bp region between *IL15* and *ZNF330*, 31,334 bp downstream of *IL15* and 1245 bp upstream of *ZNF330*, had the fourth-most significant effect of Chr17 (#26 overall) and seven other significant effects (Table 3, Figure 3b). *IL15* is a cytokine that regulates T and natural killer cell activation and proliferation [24], and *ZNF330* is predicted to enable zinc ion binding activity [25].

### 2.5. Additive Effects of Other Chromosomes

For the remaining nine chromosomes with significant additive effects, only Chr15 and Chr24 had substantial numbers of significant effects, fifteen effects for Chr15 and eleven effects for Chr24, whereas chromosomes 6, 7, 10, 19, 25, 29 and 31 each had 1–3 effects. Examples of these effects are summarized in Table 4, whereas effects not discussed in the main text can be found in Appendix A.

The fifteen effects of Chr15 were distributed over the 26–67 Mb region and the four most significant effects of Chr15 were in *SLC1A2* (#48), *IL10RA* (#89), *SPON1* (#90), and downstream of *DCDC1* ($95) covering a large distance of the 26–67 Mb region (Figure 4a). *SLC1A2* encodes a membrane-bound protein as the principal transporter that clears the excitatory neurotransmitter glutamate and glutamate clearance is necessary for proper synaptic activation and to prevent neuronal damage from excessive activation of glutamate receptors [26]. *IL10RA* encodes a protein that is a receptor for interleukin 10, and is structurally related to interferon receptors [27]. *SPON1* is predicted to be an extracellular matrix structural constituent and to be involved in cell adhesion [28]. The four most significant effects of Chr24 were in a narrow 30.38–31.01 Mb region about 0.63 Mb in size (Figure 4b), downstream of *SS18*, in *TAF4B* (2 effects) and upstream of *TAF4B* with overall rankings of #15, #18, #33 and #37, respectively, where *SS18* is involved in positive regulation of transcription by RNA polymerase II [29], and *TAF4B* is involved in initiation of transcription of genes by RNA polymerase II [30]. Chr31, the X-Y nonrecombining region of the X-chromosome, had three significant effects (Figure 4c). The top two effects of this chromosome were in *MAGED1-MAGED4B* (#73) and downstream of *NUDT11* (#75). The SNP in *MAGED1-MAGED4B* had the most negative allelic effect among all SNPs with a low allele frequency of 0.07. Chr06 had three significant effects (#104, #108 and #175 overall) in the *SLC4A4-GC-NPFFR2* region (Figure 4d), which had been reported to have highly significant effects on milk yield, fertility and somatic cell score [12,31]. Chr10 had two significant effects slightly above the log_10_(1/p) = 8 cutoff value for declaring significance, one in *RASGRP1*, which activates the Erk/MAP kinase cascade and regulates T-cells and B-cells development, homeostasis and differentiation [32], and one in *LIPC* which Enables phospholipase A1 activity and triglyceride lipase activity [33]. Chr07 had one significant effect (#78 overall) downstream of *OR7A5*. Olfactory receptors interact with odorant molecules in the nose, to initiate a neuronal response that triggers the perception of a smell. Chr19 had two significant effects, one in *ABCA10* (#96 overall) and one in *ABCA9* (#125 overall).

### 2.6. Gene Ontology Analysis

Gene Ontology (GO) analysis was conducted to understand the potential biological functions of candidate coding genes of the 200 significant additive effects, and the results are summarized in Appendix A. The GO results provided many more details about the biological functions of the candidate genes than described thus far in this article, e.g., *ESR1* and *KITLG* each had over one hundred biological functions (Appendix A). The GO results also identified a few genes involved in embryonic placenta development, including *KRT8* of Chr05 downstream of the #17 effect (Table 2), *EDNRA* of Chr17 upstream of the #61 effect, *GCM1* of Chr23 upstream of the #157 and #168 effects, and *MAP3K4* of Chr09 with the #155 effect (Appendix A). However, the GO results did not include some of the information we collected from journal articles and the National Center for Biotechnology Information (NCBI), e.g., the *ESR1* fusions, the in utero biological functions of *KITLG*, and the association of *PKHD1* and *ACOT13* with kidney disease. Although the GO analysis identified over 2000 biological functions of the candidate genes (Appendix A), none of those biological functions was identified to have a direct effect on retained placenta. In contrast, the GWAS results of this study provided Holstein-specific and high-confidence evidence for the potential associations between the candidate genes and RETP. The combination of the GWAS results of this study with the GO results as well as the biological information of the candidate genes collected elsewhere should provide useful functional annotations of the candidate genes and indications of the potential genetic mechanisms of the significant SNP effects affecting RETL in Holstein cows.

## 3. Materials and Methods

### 3.1. Holstein Population and SNP Data

The Holstein population in this study had 632,212 cows with RETP phenotypic observations and 78,964 original and imputed SNPs. With the requirement of 0.05 minor allele frequency, 74,747 SNPs were used in the GWAS analysis. The SNP positions were those from the ARS-UCD1.3 cattle genome assembly. Genes containing or in proximity to highly significant SNP effects were identified as candidate genes affecting RETP. The RETP phenotypic values used in the GWAS analysis were the phenotypic residuals after removing fixed nongenetic effects available from the December 2023 U.S. Holstein genomic evaluation data.

### 3.2. GWAS Analysis

The GWAS analysis used an approximate generalized least squares (AGLS) method. The AGLS method combines the least squares (LS) tests implemented by EPISNP1mpi [34,35] with the estimated breeding values from a routine genetic evaluation using the entire U.S. Holstein population. The statistical model was:(1)$$y=\mu I+X_{g}g+Za+e=Xb+Za+e$$
where **y** = column vector of phenotypic deviation after removing fixed nongenetic effects such as heard-year-season (termed as ‘yield deviation’ for any trait) using a standard procedure for the CDCB/USDA genetic and genomic evaluation; µ = common mean; **I** = identity matrix; **g** = column vector of genotypic values; $X_{g}$ = model matrix of **g**; $b={\left(\mu ,\;g^{\prime }\right)}^{\prime }$, $X=(I,\;X_{g})$; **a** = column vector of additive polygenic values; **Z** = model matrix of **a**; and **e** = column vector of random residuals. The first and second moments of Equation (1) are: E(y)=Xb and $\mathrm{var}(y)=V={ZGZ}^{\prime }+R=\sigma_{a}^{2}{ZAZ}^{\prime }+\sigma_{e}^{2}I$, where $\sigma_{a}^{2}$ = additive variance, **A** = additive relationship matrix, and $\sigma_{e}^{2}$ = residual variance. The problem of estimating the **b** vector that includes SNP genotypic values in Equation (1) is the requirement of inverting the **V** if the generalized least squares (GLS) method is used, or solving the mixed model equations (MME) [36]. Either the GLS or MME method for each of the genome-wide SNPs is computationally demanding for our sample size. To avoid these computing difficulties, the GWAS used the method of approximate GLS (AGLS) that replaces the polygenic additive values (**a**) with the best linear unbiased prediction based on pedigree relationships [12,21,31,37]. The significance tests for additive and dominance SNP effects used the *t*-tests of the additive and dominance contrasts of the estimated SNP genotypic values [34,38]. The t-statistic of the AGLS was calculated as:(2)$$t_{j}=\frac{{|L}_{j}|}{\sqrt{{\mathrm{var}(L}_{j})}}=\frac{\left|s_{j}\hat{g}\right|}{v\sqrt{s_{j}{\left(X^{\prime }X\right)}_{\mathrm{gg}}^{-}s_{j}^{\prime }}},\;\;j=a,d$$
where $L_{j}$ = additive or dominance contrast; $\sqrt{{\mathrm{var}(L}_{j})}$ = standard deviation of the additive or dominance contrast; $s_{a}$ = row vector of additive contrast coefficients = $\left[\begin{array}{ccc} P_{11}{/p}_{1} & 0{.5P}_{12}{(p}_{2}-p_{1})/(p_{1}p_{2}) & -P_{22}{/p}_{2} \end{array}\right]$; $s_{d}$ = row vector of dominance contrast coefficients = $\left[\begin{array}{ccc} -0.5 & 1 & 0.5 \end{array}\right]$; $v^{2}=\left(y-X\hat{b}{)}^{\prime }(y-X\hat{b}\right)/(n-k)$ = estimated residual variance; $\hat{g}$ = column vector of the AGLS estimates of the three SNP genotypic effects of $g_{11}$, $g_{12}$, and $g_{22}$ from Equation (4); ${\left(X^{\prime }X\right)}_{\mathrm{gg}}^{-}$ = submatrix of ${\left(X^{\prime }X\right)}^{-}$ corresponding to $\hat{g}$; and where $p_{1}$ = frequency of $A_{1}$ allele, $p_{2}$ = frequency of $A_{2}$ allele of the SNP, $P_{11}$ = frequency of $A_{1}A_{1}$ genotype, $P_{12}$ = frequency of $A_{1}A_{2}$ genotype, $P_{22}$ = frequency of $A_{2}A_{2}$ genotype, n = number of observations and k = rank of **X**.

Additive effects of each SNP were estimated using three measures, the average effect of gene substitution, allelic mean, and allelic effect of each allele based on quantitative genetics definitions [38,39]. The allelic mean ($\mu_{i}$), the population mean of all genotypic values of the SNP (μ), the allelic effect ($a_{i}$), and the average effect of gene substitution of the SNP (α) are:(3)$$\mu_{1}=P_{11.1}g_{11}+\;0{.5P}_{12.1}g_{12}$$
(4)$$\mu_{2}=0{.5P}_{12.2}g_{12}+\;P_{22.2}g_{22}$$
(5)$$\mu =\sum_{i=1}^{2}p_{i}\mu_{i}$$
(6)$$a_{i}\;=\;\mu_{i}-\mu ,i=1,2$$

(7)$$\alpha \;\;=\;L_{a}=s_{a}\hat{g}\;=\;a_{1}-a_{2}=\;\;\mu_{1}-\mu_{2}$$
where $P_{11.1}=P_{11}{/p}_{1}$, $P_{12.1}=P_{12}{/p}_{1}$, $P_{12.2}=P_{12}{/p}_{2}$, and $P_{22.2}=P_{22}{/p}_{2}$. The additive effect measured by the average effect of gene substitution of Equation (7) is the distance between the two allelic means or effects of the same SNP, and is the fundamental measure for detecting SNP additive effects as shown by the t-statistic of Equation (2). However, the allelic effect of Equation (7) is not comparable across SNPs because the allelic effect is affected by the genotypic mean of the SNP defined by Equation (6). To compare allelic effects across SNPs, we replaced the SNP genotypic mean (μ) in Equation (6) with the average of all SNP genotypic means ($\mu_{\mathrm{all}}$):(8)$$a_{i}\;=\;\mu_{i}-\mu_{\mathrm{all}},i=1,2$$Equation (8) was used only for the purpose of graphical display of allelic effects.

### 3.3. Gene Ontology (GO) Analysis

To understand the potential functions of selected candidate genes, the Gene Ontology (GO) analysis was performed using the OmicShare platform (www.omicshare.com/tools, accessed on 15 May 2024).

## 4. Conclusions

The GWAS results in this study indicated that RETP in U.S. Holstein cows was affected by multiple genetic variants with additive effects. Although the exact genetic mechanism underlying RETP remained unknown, these significant effects involved genes with a variety of biological functions reported elsewhere including *ESR1* gene fusions, immunity, genetic effects on fertility, health and milk production, kidney disease, lactation, *KIT* ligand in utero, and basal body and cilia formation. The SNP effects detected in this study along with known biological functions of genes with or near the SNP effects provided a new understanding of genetic factors underlying RETP in U.S. Holstein cows and provided comparative information about the genetic mechanism of retained placenta in other species.

## Figures and Tables

**Figure 1 ijms-25-05551-f001:**
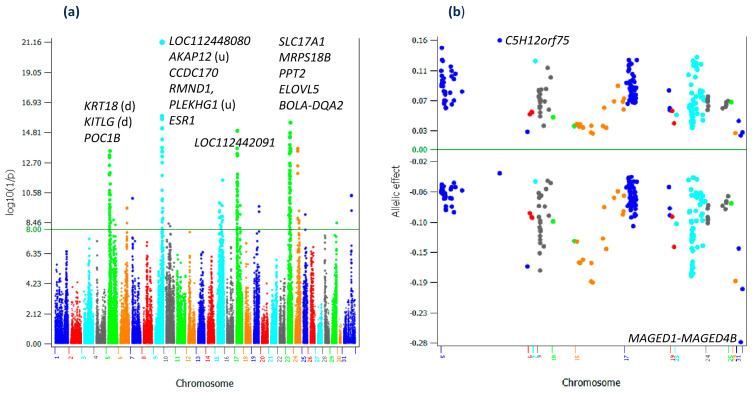
Graphical view of additive effects. (**a**) Manhattan plot of additive effects of all chromosomes. (**b**) allelic effects of the 151 significant SNPs. ‘u’ indicates the SNP is upstream of the gene. ‘d’ indicates the SNP is downstream of the gene.

**Figure 2 ijms-25-05551-f002:**
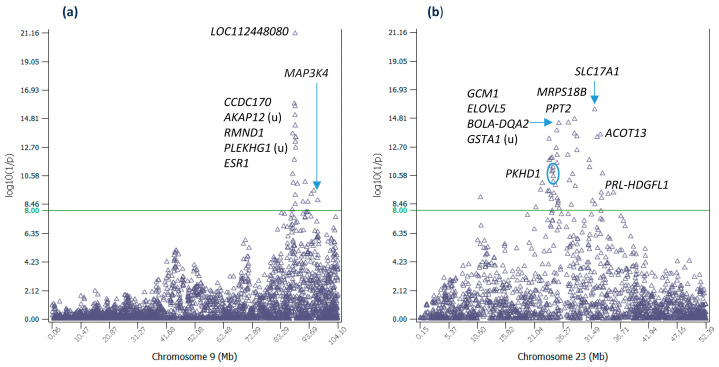
Additive effects of Chr09 and Chr23. (**a**) Statistical significance of additive effects of Chr09 SNPs. (**b**) Statistical significance of additive effects of Chr23 SNPs. ‘u’ indicates the SNP is upstream of the gene.

**Figure 3 ijms-25-05551-f003:**
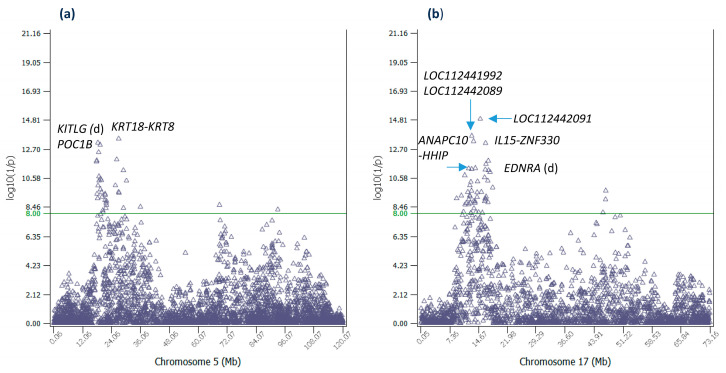
Additive effects of Chr05 and Chr17. (**a**) Statistical significance of additive effects of Chr05 SNPs. (**b**) Statistical significance of additive effects of Chr17 SNPs. ‘d’ indicates the SNP is downstream of the gene.

**Figure 4 ijms-25-05551-f004:**
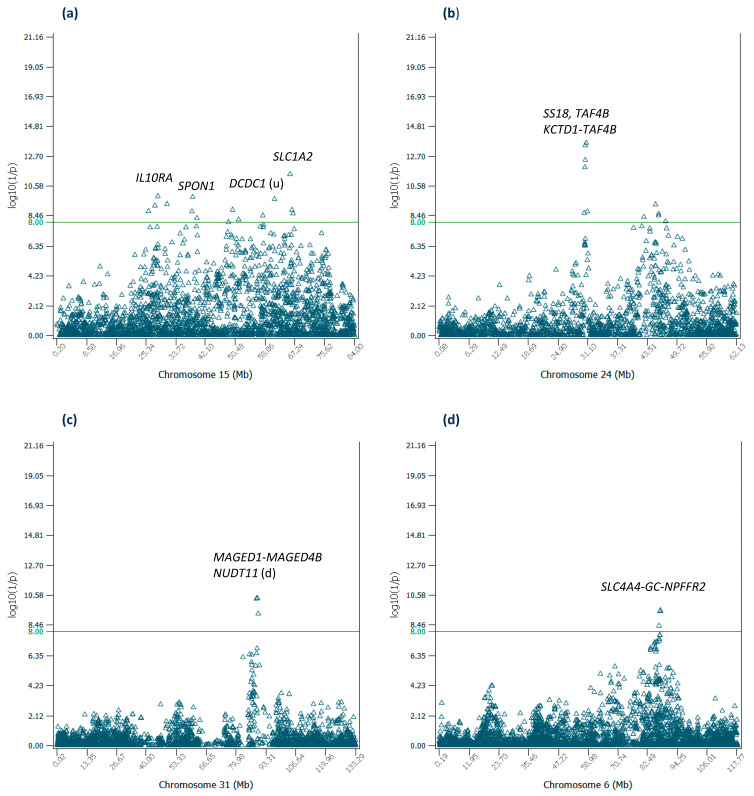
Additive effects of Chr15, Chr24, Chr31 and Chr06. (**a**) Statistical significance of additive effects of Chr15 SNPs. (**b**) Statistical significance of additive effects of Chr24 SNPs. (**c**) Statistical significance of additive effects of Chr31 SNPs. (**d**) Statistical significance of additive effects of Chr06 SNPs. ‘u’ indicates the SNP is upstream of the gene. ‘d’ indicates the SNP is downstream of the gene.

**Table 1 ijms-25-05551-t001:** Significant additive effects of Chr09 for RETP.

SNP	Position	Candidate Gene	Effect	Log_10_(1/p)	al+	ae+	f_al+	al−	ae−	f_al−
*rs42026926*	87610236	*PLEKHG1* (u)	0.183	13.74	1	0.080	0.564	2	−0.103	0.436
*rs42026914*	87634273	*PLEKHG1* (u)	0.158	10.75	1	0.075	0.526	2	−0.0833	0.474
*rs43615609*	88220932	*MTHFD1L* (d)	0.175	11.80	1	0.062	0.644	2	−0.112	0.356
*rs43615544*	88254314	*AKAP12* (u)	−0.210	15.96	2	0.062	0.706	1	−0.148	0.294
*rs43612983*	88482928	*RMND1*	0.197	15.10	1	0.070	0.643	2	−0.127	0.357
*rs43611719*	88519048	*LOC112448080*	−0.246	21.16	2	0.071	0.71	1	−0.175	0.29
*rs43611710*	88529494	*CCDC170*	−0.199	15.76	2	0.070	0.646	1	−0.128	0.354
*rs43611701*	88540232	*CCDC170*	−0.182	14.33	2	0.081	0.556	1	−0.102	0.444
*rs43610539*	88598336	*CCDC170*	−0.189	13.13	2	0.057	0.699	1	−0.132	0.301
*rs43608567*	88684552	*ESR1*	−0.177	12.70	2	0.063	0.642	1	−0.114	0.358
*rs3423297865*	88722921	*ESR1*	0.188	13.46	1	0.068	0.641	2	−0.12	0.359
*rs43767108*	88739977	*ESR1*	0.188	13.48	1	0.068	0.641	2	−0.121	0.359

‘u’ indicates the SNP is upstream of the gene. ‘d’ indicates the SNP is downstream of the gene. ‘effect’ is the additive effect of the SNP as the difference between allelic effects of ‘allele 1’ and ‘allele 2’ (Equation (6)). ‘al+’ is the positive allele, ‘al–’ is the negative allele, ‘ae+’ is the allelic effect of the positive allele, and ‘ae−’ is the allelic effect of the negative allele (Equation (7)). ‘f_al+’ is the frequency of the positive allele. ‘f_al−’ is the frequency of the negative allele.

**Table 2 ijms-25-05551-t002:** Top-20 significant additive effects of Chr23 for RETP.

SNP	Position	Candidate Gene	Effect	Log_10_(1/p)	al+	ae+	f_al+	al−	ae−	f_al−
*rs110556135*	23704336	*PKHD1* (u)	0.224	13.33	1	0.042	0.813	2	−0.182	0.187
*rs43561755*	23750046	*PKHD1* (u)	−0.216	11.75	2	0.039	0.819	1	−0.177	0.181
*BTA-81662-no-rs*	23890370	*PKHD1* (u)	−0.166	11.30	2	0.062	0.629	1	−0.104	0.371
*rs41670209*	24060064	*PKHD1* (u)	−0.183	11.88	2	0.052	0.718	1	−0.131	0.282
*rs136181786*	24252119	*PKHD1*	−0.180	11.94	2	0.058	0.68	1	−0.123	0.32
*rs137762108*	25024407	*LOC112443711-LOC112443751*	−0.199	12.65	2	0.051	0.746	1	−0.148	0.254
*rs135832378*	25119540	*GSTA1* (u)	0.186	13.95	1	0.073	0.609	2	−0.113	0.391
*rs110144575*	25142236	*GSTA5*	0.218	11.54	1	0.039	0.821	2	−0.179	0.179
*rs110043199*	25175265	*LOC112443730*	0.219	12.14	1	0.039	0.82	2	−0.180	0.18
*rs135146076*	25491332	*ELOVL5, BOLA-DQA2*	−0.190	14.48	2	0.067	0.645	1	−0.123	0.355
*rs133177329*	27122907	*ENSBTAG00000048304*	0.172	12.15	1	0.105	0.388	2	−0.067	0.612
*rs110358203*	27231641	*PPT2*	−0.192	14.51	2	0.124	0.353	1	−0.068	0.647
*rs110277462*	28345983	*DHX16*	0.186	13.72	1	0.121	0.351	2	−0.065	0.649
*rs3423504515*	28367574	*MRPS18B*	0.193	14.78	1	0.126	0.35	2	−0.068	0.65
*rs3423514031*	28777609	*LOC785873*	0.183	13.50	1	0.115	0.373	2	−0.068	0.627
*rs3423494105*	28783122	*TRIM26*	0.193	12.48	1	0.051	0.737	2	−0.142	0.263
*rs3423498878*	31063056	*TRNAF-GAA_18,* *TRNAI-UAU_6*	−0.165	11.75	2	0.085	0.486	1	−0.080	0.514
*rs109821904*	32057953	*SLC17A1*	−0.196	15.50	2	0.125	0.363	1	−0.071	0.637
*rs134698463*	32498379	*CARMIL1*	−0.178	13.46	2	0.086	0.514	1	−0.091	0.486
*rs3423509404*	33077628	*ACOT13*	−0.177	13.65	2	0.085	0.517	1	−0.091	0.483

‘u’ indicates the SNP is upstream of the gene. ‘effect’ is the additive effect of the SNP as the difference between allelic effects of ‘allele 1’ and ‘allele 2’ (Equation (6)). ‘al+’ is the positive allele, ‘al–‘ is the negative allele, ‘ae+’ is the allelic effect of the positive allele, and ‘ae−’ is the allelic effect of the negative allele (Equation (7)). ‘f_al+’ is the frequency of the positive allele. ‘f_al−’ is the frequency of the negative allele.

**Table 3 ijms-25-05551-t003:** Significant additive effects of Chr05 and Chr17 for RETP.

SNP	Chr	Position	Candidate Gene	Effect	Log_10_(1/p)	al+	ae+	f_al+	al−	ae−	f_al−
*rs110165899*	5	27005657	*KRT18-KRT8*	0.184	13.51	1	0.118	0.358	2	−0.066	0.642
*rs41587994*	5	18367350	*KITLG* (d)	0.198	13.22	1	0.147	0.258	2	−0.051	0.742
*rs41603721*	5	19295836	*POC1B*	−0.187	13.06	2	0.129	0.307	1	−0.057	0.693
*rs29012239*	5	18679362	*LOC104972350-* *LOC104972370*	0.185	12.51	1	0.131	0.292	2	−0.054	0.708
*rs135127542*	5	26178969	*CALCOCO1* (u)	−0.178	11.99	2	0.122	0.313	1	−0.056	0.687
*rs136124246*	5	17909902	*CEP290*	0.176	11.91	1	0.117	0.332	2	−0.058	0.668
*rs110506590*	5	17780338	*C5H12orf50* (u)	0.175	11.84	1	0.117	0.332	2	−0.058	0.668
*rs137107793*	5	28970729	*HIGD1C*	0.162	11.19	1	0.070	0.564	2	−0.091	0.436
*rs137455368*	5	18799786	*LOC104972350-* *LOC104972370*	0.161	10.75	1	0.101	0.373	2	−0.060	0.627
*rs109747382*	5	19468540	*ATP2B1* (u)	−0.159	10.58	2	0.099	0.376	1	−0.060	0.624
*rs110739449*	17	15056547	*LOC112442091*	−0.187	14.94	2	0.079	0.575	1	−0.107	0.425
*rs135912416*	17	12902838	*LOC112441992-* *LOC112442089*	−0.180	13.70	2	0.092	0.487	1	−0.088	0.513
*rs137219013*	17	13350634	*ANAPC10-HHIP*	0.176	13.31	1	0.078	0.559	2	−0.098	0.441
*rs109572161*	17	16399921	*IL15-ZNF330*	−0.174	13.18	2	0.079	0.546	1	−0.095	0.454
*rs109486788*	17	17090817	*TBC1D9*	0.169	11.88	1	0.077	0.543	2	−0.092	0.457
*rs108973145*	17	16489833	*IL15-ZNF330*	−0.164	11.64	2	0.072	0.561	1	−0.092	0.439
*rs137504512*	17	13801294	*TRNAC-GCA_189,* *TRNAG-UCC_41*	−0.163	11.34	2	0.081	0.503	1	−0.082	0.497
*rs41838712*	17	12116917	*REELD1*	0.168	11.32	1	0.102	0.394	2	−0.066	0.606
*rs41599601*	17	12866447	*LOC112441992*	0.160	11.28	1	0.075	0.531	2	−0.085	0.469

‘u’ indicates the SNP is upstream of the gene. ‘d’ indicates the SNP is downstream of the gene. ‘effect’ is the additive effect of the SNP as the difference between allelic effects of ‘allele 1’ and ‘allele 2’ (Equation (6)). ‘al+’ is the positive allele, ‘al–’ is the negative allele, ‘ae+’ is the allelic effect of the positive allele, and ‘ae−’ is the allelic effect of the negative allele (Equation (7)). ‘f_al+’ is the frequency of the positive allele. ‘f_al−’ is the frequency of the negative allele.

**Table 4 ijms-25-05551-t004:** Significant additive effects of seven selected chromosomes for RETP.

SNP	Chr	Position	Candidate Gene	Effect	Log_10_(1/p)	al+	ae+	f_al+	al−	ae−	f_al−
*rs109034709*	6	87316810	*NPFFR2*	−0.152	9.53	2	0.057	0.628	1	−0.096	0.372
*rs110434046*	6	87184768	*GC-NPFFR2*	−0.152	9.47	2	0.056	0.628	1	−0.095	0.372
*rs137664040*	6	86795926	*SLC4A4*	−0.143	8.45	2	0.057	0.603	1	−0.086	0.397
*rs3423224824*	7	8274451	*OR7A5* (d)	−0.174	10.18	2	0.128	0.264	1	−0.046	0.736
*rs109718130*	10	34176744	*RASGRP1*	−0.150	8.41	2	0.047	0.689	1	−0.104	0.311
*rs43626966*	10	51788612	*LIPC*	0.166	8.23	1	0.036	0.785	2	−0.130	0.215
*rs133296429*	15	65887087	*SLC1A2*	0.163	11.46	1	0.072	0.56	2	−0.091	0.440
*rs110222319*	15	28670668	*IL10RA*	0.200	9.88	1	0.035	0.827	2	−0.165	0.173
*rs133481154*	15	38458720	*SPON1*	0.198	9.83	1	0.035	0.823	2	−0.163	0.177
*rs110235930*	19	61387218	*ABCA10*	−0.153	9.62	2	0.057	0.625	1	−0.096	0.375
*rs41932313*	19	61518347	*ABCA9*	−0.179	9.24	2	0.038	0.788	1	−0.141	0.212
*rs43772736*	24	30783746	*SS18* (d)	0.178	13.68	1	0.078	0.559	2	−0.099	0.441
*rs136103342*	24	30564828	*TAF4B*	−0.179	13.51	2	0.076	0.579	1	−0.104	0.421
*rs207730478*	24	30578431	*TAF4B*	−0.170	12.47	2	0.072	0.579	1	−0.099	0.421
*rs110190049*	24	30486009	*KCTD1-TAF4B*	−0.167	11.95	2	0.070	0.579	1	−0.097	0.421
*rs133376988*	31	89494444	*MAGED1-MAGED4B*	0.299	10.41	1	0.022	0.927	2	−0.277	0.073
*rs136268223*	31	89068436	*NUDT11* (d)	−0.184	10.36	2	0.042	0.772	1	−0.142	0.228
*rs137683400*	31	89839370	*LOC100297099*	−0.226	9.32	2	0.027	0.883	1	−0.200	0.117

‘d’ indicates the SNP is downstream of the gene. ‘effect’ is the additive effect of the SNP as the difference between allelic effects of ‘allele 1’ and ‘allele 2’ (Equation (6)). ‘al+’ is the positive allele, ‘al–’ is the negative allele, ‘ae+’ is the allelic effect of the positive allele, and ‘ae−’ is the allelic effect of the negative allele (Equation (7)). ‘f_al+’ is the frequency of the positive allele. ‘f_al−’ is the frequency of the negative allele.

## Data Availability

The original genotype data are owned by third parties and maintained by the Council on Dairy Cattle Breeding (CDCB). A request to CDCB is necessary for getting data access on research, which may be sent to: João Dürr, CDCB Chief Executive Officer (joao.durr@cdcb.us). All other relevant data are available in the manuscript and Appendix A.

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
