# Peer review of "Large-Sample Genome-Wide Association Study of Resistance to Retained Placenta in U.S. Holstein Cows"

_ijms, 2024, doi:10.3390/ijms25105551_

Round 1
Reviewer 1 Report
Comments and Suggestions for Authors
Sincerely Author,
GWAS studies of functional/ health traits are awaited in research and expert communities with high relevance as promising future solutions to the main economical problems in dairy industry of today.
Therefore, I have read your submission with high interest.
Generally, I was positively surprised by Introduction part of your manuscript as it is written in concise and informative manner. However, even believing that you used proper/ relevant references, I think that study books like Falconers adn Mackays Introduction to quantitative genetics is to broad and general. I can understand that this is the kind of "bible" in this particular area but in this paticular case, indicate pages you are referencing to. Same is valid in case of Henderson's reference as this is basic work in implementation of linear models, however I believe that it is mentioned as well in the Falconers "bible". But as written earlier, I can understand authors point with referencing those.
With this comment it lead to my second point that Authors coment use of standard CDCB/ USDA methodology is not appropriate and as minimum valid reference to the document with description of this "Standard" methodology should be included. Otherwise, I don't thing that it is a standard methodology - this description is bit trivial.
My comment to the Introduction and Methodology are anyway not decreasing value of this submission and I generally agree with it. However, I would like authors to revise those particular parts and upgrade them.
Regarding Results and Discussion several questions/ comments came to my mind and I would like to ask authors to react on them and adopt if relevant regarding their work:
1) When mentioning several allellic effects per chromosome, did you consider that they could act in linkage?
2) If so, pathaway analysis could clarify the allelic cascade
3) I am missing also more detailed biological description i.e. gene onthology
4) I am surprised that you were able to estimate "only" additive effects because in case of Holstein and high homozygosity in their genome, dominance effect would be expected
5) I.e. is there only possitive/ directional selection presure? no effect of stabiling-centripetal selection in case of Holstein?
Regarding issues of selection, I would expect that authors dedicate some paragraph in explaing effect of selection pressure in the Holstein genome, regarding the results.
Author Response
1) When mentioning several allellic effects per chromosome, did you consider that they could act in linkage?
ANSWER: This is a single-SNP GWAS. The significant effects should have different degrees of LD, stronger LD for two loci close to each other, and the LD becomes weaker as the loci are separated by larger distances.
2) If so, pathaway analysis could clarify the allelic cascade
ANSWER: Many diagrams are available showing relationships among candidate genes. We are not presenting any of those.
3) I am missing also more detailed biological description i.e. gene onthology
ANSWER: This revision adds GO results and the text is revised accordingly.
4) I am surprised that you were able to estimate "only" additive effects because in case of Holstein and high homozygosity in their genome, dominance effect would be expected
ANSWER: Dominance effects are trait specific. We have reported dominance effects for production traits (ref #30) and reproduction traits (ref #11 and #20) in Holstein cows.
5) I.e. is there only possitive/ directional selection presure? no effect of stabiling-centripetal selection in case of Holstein?
ANSWER: Selection response is negligible, because genomic evaluation of RETP only started in 2018.
Regarding issues of selection, I would expect that authors dedicate some paragraph in explaing effect of selection pressure in the Holstein genome, regarding the results.
ANSWER: Systematic selection for RETP did not exist until this trait was included in the ‘net merit’ selection index. Selection response is negligible because genomic evaluation of RETP only started in 2018. RETP was one of the six health traits in the index and collectively the six health traits only had 1.7% relative emphasis. Sorting out any selection response for RETP should require a separate study.
Reviewer 2 Report
Comments and Suggestions for Authors
This manuscript by Prakapenka et al. investigates a large-sample genome-wide association study of resistance to retained placenta (RETP) in U.S. Holstein cows. The study's findings, based on a comprehensive data analysis, are credible and provide new insights into the genetic underpinnings of RETP in this cattle breed. However, further revisions are needed to address some issues.
All abbreviated terms in the abstract should be spelled out in full.
In table annotations and legends, each abbreviation must be explained in full.
The study's conclusions are drawn solely from large-scale data analysis. Further investigation is required to validate their effectiveness in addressing resistance to retained placenta. How did the author consider it?
Additionally, a concise description of the Holstein population involved in the study should be included.
The background description is relatively limited, and it is necessary to add some content in the introduction section.
Author Response
All abbreviated terms in the abstract should be spelled out in full. In table annotations and legends, each abbreviation must be explained in full.
ANSWER: Full gene names are now provided in Table S2.
The study's conclusions are drawn solely from large-scale data analysis. Further investigation is required to validate their effectiveness in addressing resistance to retained placenta. How did the author consider it?
ANSWER: Full gene names are now provided in Table S2.
Additionally, a concise description of the Holstein population involved in the study should be included.
ANSWER: The description of the Holstein population is in Section 3.1.
The background description is relatively limited, and it is necessary to add some content in the introduction section.
ANSWER: The lack of genetic studies on the same trait was the reason for the short introduction. There were a few other genetic studies of a much smaller scale or for a different breed with questionable results. A concise introduction should be better than a lengthy one with questionable and irrelevant results. Based on this principle, this revision only adds a review article, ref #1.